# Exposure to Public Open Spaces and Leisure-Time Physical Activity: An Analysis of Adults in Primary Health Care in Brazil

**DOI:** 10.3390/ijerph19148355

**Published:** 2022-07-08

**Authors:** Rafael Luciano de Mello, Adalberto Aparecido dos Santos Lopes, Rogério César Fermino

**Affiliations:** 1Research Group on Environment, Physical Activity and Health, Federal University of Technology-Paraná, Curitiba 81310-900, Brazil; rf.mello1@gmail.com; 2Postgraduate Program in Physical Education, Federal University of Technology-Paraná, Curitiba 81310-900, Brazil; 3Research and Study Group in Urban Environment and Health, Federal University of Santa Catarina, Florianopolis 88040-900, Brazil; aadalberto@hotmail.com; 4Observatory for Urban Health in Belo Horizonte, Federal University of Minas Gerais, Belo Horizonte 30130-100, Brazil; 5Postgraduate Program in Physical Education, Federal University of Paraná, Curitiba 81531-980, Brazil

**Keywords:** leisure activities, parks, built environment, evaluation of health programs and projects, public policy

## Abstract

Public open spaces (POS) enable many leisure activities. However, few studies have evaluated how different exposures to POS with structured physical activities (PA) could affect the leisure-time PA (LTPA) of adults with chronic diseases. This study aimed to analyze the associations between different exposures to POS and LTPA in adults in primary health care in Brazil. In 2019, a cross-sectional study was conducted with a representative sample of 615 users of 15 Basic Health Units in São José dos Pinhais. The POS exposure was measured by the presence and distance (GIS), and the usage and frequency of usage were self-reported. The IPAQ identified walking and moderate-to-vigorous PA (MVPA). Data were analyzed with multilevel Poisson regression adjusted for the possible confounders (sex, age, economic level, chronic diseases). The prevalence of walking and MVPA ≥150 min/week was 12% and 14%, respectively. The usage of POS increased by 52% the likelihood of walking ≥10 min/week (PR: 1.52; 95%CI: 1.02–2.21), while usage ≥2 times/week increased by 2 times the likelihood of this outcome (PR: 2.01; 95%CI: 1.28–3.05). The usage of POS increased by 2.7 times the likelihood of MVPA ≥10 min/week (PR: 2.73; 95%CI: 1.86–3.97), and usage ≥2 times/week increased by 3.5 times the likelihood of this outcome (PR: 3.45; 95%CI: 2.14–5.42). Finally, the usage of POS increased by 3 times the likelihood of MVPA ≥150 min/week (PR: 3.03; 95%CI: 1.92–4.74), while usage ≥2 times/week increased by 3.8 times the likelihood of this outcome (PR: 3.81; 95%CI: 2.19–6.38). In conclusion, only the usage of POS is positively associated with walking and MVPA in the leisure time of adult primary health care users.

## 1. Introduction

Despite governments’ efforts and investments to promote physical activity (PA) at the population level, the overall prevalence of insufficient PA is high, and 28% of adults do not meet the PA recommendations [1,2,3]. This figure is even higher in Latin America (39%), especially in Brazil (47%) [1], posing a burden to the public health system with increasing hospitalizations and general health care costs as well as contributing to global premature mortality [4,5,6,7].

The Brazilian public health system, Sistema Único de Saúde (SUS; Unified Health System) is based on the principles of universality, integrality, and social participation and has been recognized as a citizen’s right since the Brazilian constitution implemented the system in 1988; it covers from outpatient care to highly complex procedures such as hemodialysis and organ transplantation [8]. It is composed of different health establishments, specifically more than 42,000 Basic Health Units (BHUs), public health centers with the purpose of serving up to 80% of the population’s problems without needing a hospital, and in most cases the first primary health care contact [8,9,10]. It is estimated that 50% of the Brazilian population (≈110 million people) are registered in primary health care, and ≈11% of adults used some service just in 2019 [9]. Primary health care users present different characteristics than the overall population, such as lower socioeconomic status and a higher prevalence of chronic diseases [8,9,10]. This profile may partially explain the higher prevalence of physical inactivity in this population [10,11,12]. For example, studies conducted in Southern Brazil showed a leisure-time PA (LTPA) prevalence of ≈23% in BHU users [13,14] compared with 33% in a population-based survey [15].

Lower socioeconomic status and a diagnosis of a chronic diseases are important determinants that are inversely associated with LTPA [11,12]. In addition to individual characteristics, PA is determined by factors on interpersonal, environmental, political, and global levels [11,12]. At the environmental level, much recent evidence from systematic reviews has shown that exposure to public open spaces (POS) may facilitate engagement in PA, especially in the leisure domain [11,12,16,17,18,19,20,21]. However, most studies have explored the associations between exposure to parks or green areas and PA, which does not include the diversity of locations that could contribute to LTPA [16,22,23,24]. The POS context, type, and structure may affect the population’s involvement in different types of LTPA (walking, running, sports), which makes relevant the analysis of each location separately [16,25]. For example, parks and beaches may contribute to walking [22,23], while sports and recreational centers may be a better predictor of moderate-to-vigorous PA (MVPA) [15,23].

However, there is an inequality in POS spatial distribution and quality [26,27,28,29,30]. In Brazil, for example, plazas are the most common POS and contain some PA features (playgrounds, fitness zones, sports courts) [15,28,31,32,33]. However, most of them do not offer community-based PA programs that provide qualified staff and supervised activities to the local population, such as those found in some Brazilian cities [18,32,34,35,36]. These programs provide access to LTPA, mainly among more vulnerable and less exposed to PA population groups like women, low-income citizens, and those with chronic diseases [35,36], which is a similar profile to the primary health care users [9,10].

Most of the evidence available has evaluated the association between exposure to parks or green areas (presence and distance) and LTPA separately [11,12,17,19,21]. No studies have examined the association between broad exposure to POS areas (presence, distance, usage, and frequency of usage) through a community-based PA program and LTPA of adults in primary health care in Brazil. This gap does not allow for extrapolating the results from other studies to this population subgroup [9,10]. Exploring this association is essential for guiding strategies and supporting the actions of policymakers and health professionals to promote PA in primary health care [37,38]. Therefore, this study aimed to analyze the associations between different exposures to POS and LTPA in adults in primary health care in Brazil.

## 2. Materials and Methods

The description of information followed the requirements suggested by Strengthening the Reporting of Observational Studies in Epidemiology (STROBE) [39].

### 2.1. Study Characteristics, Local Contextualization, and Ethical Aspects

In 2019, a quantitative, observational, and cross-sectional study was conducted with a representative sample of adults in BHUs in urban areas in São José dos Pinhais (Paraná state, Southern Brazil). The project’s general aim was to evaluate PA and sedentary behavior counseling, LTPA, and some of their determinants [13,34]. The BHUs are public health centers strategically distributed in cities that offer free access to primary health care by physicians, nurses, pharmacists, physiotherapists, nutritionists, psychologists, and community health agents [9].

São José dos Pinhais is a medium-sized developed city that comprises 323,000 inhabitants, located in the metropolitan region of Curitiba (capital of Paraná state), 19 km from downtown to downtown. The Human Development Index (0.758) and the Gini index per capita (0.459) are high [40]. The city is at 906 m above sea level, has a subtropical climate, and has an average temperature of 20 °C (68 °F). The geographical area is 946 km^2^ (79% rural), 90% of the residents live in an urban area, and the population density is 354 inhab./km^2^ [41]. This density is higher than other medium-sized cities (e.g., Pelotas: 204 inhab./km^2^) and lower than urban centers in Brazil (e.g., Curitiba: 4027 inhab./km^2^; Rio de Janeiro: 5266 inhab./km^2^; São Paulo 7398 inhab./km^2^) [42] or other Latin America cities (e.g., Mexico City, Mexico: 6163 inhab./km^2^, Bogotá, Colombia: 4567 inhab./km^2^) [43,44].The urban area of São José dos Pinhais comprises 42 neighborhoods, 271 census tracts, and 115 POS unequally distributed in the city [29]. The city has 413 health establishments and 27 BHUs (56% in the urban area). However, only BHUs in the urban area (*n* = 15) were intentionally selected for this study, as these are accessible to 90% of the city’s population.

The study was approved by the National Commission for Ethics in Research (CONEP) of the National Health Council, with a Certificate of Presentation of Ethical Appreciation under protocol number 95985118.0.0000.0020 and by the Research Ethics Committee of the Pontifical Catholic University of Paraná under number 2882.260. Participants were consulted and informed about voluntariness and agreed to participate in the research by signing an informed consent form, according to the recommendations of CONEP.

### 2.2. Sample Size, Number of Participants and Sample Power

The number of participants was estimated based on the average number of visits to each BHU between January and February 2019 (*n* = 34,275) [13]. The sample calculation was conducted to represent the population considering a prevalence of PA counseling received through the primary health care services of 30% (identified in the literature review) [14], 95% confidence interval (CI); sampling error of four percentage points, and design effect of 1.5 [45]. As a result, the minimum number of participants was estimated at 745 people. However, with a 10% increase in losses and refusals, we estimated the need to approach 820 participants. However, we decided to approach a surplus of 100 people (*n* = 920) to reduce the estimation errors for multivariate analyses in future studies. The sample size was proportionally calculated by the number of visits to each BHU and varied from 31 to 92 users.

As this study used secondary data from a major project, the number of participants required to represent the population was estimated a posteriori. Specifically, we considered a prevalence of 21% of total LTPA (≥150 min/week) in adults who visited a BHU in southern Brazil [14], 95% confidence interval, sampling error of four percentage points, and design effect of 1.5. Therefore, the minimum sample required was 591 adults. The analytic sample was 615 adults (163 men and 452 women). A posteriori statistical analysis, conducted in G*Power software (University of Duesseldorf, Duesseldorf, Germany) [46], showed an average power of 92% (β = 20%) and a confidence interval of 95% (α = 5%).

### 2.3. Selection of Participants

The participants were selected based on their positions in their BHU waiting room, counted from one to five, from left to right starting at the BHU door. The third user was approached and invited to participate in the study [14], but in case of refusal or if the participant did not meet the inclusion criteria, the first person on the left was selected.

Only adults (≥18 years old) were eligible and invited. Among them, we excluded people who lived outside the urban area and were using the BHU for the first time, had some physical limitation for PA practice (e.g., wheelchair and crutch users), or had some cognitive or phonation limitation that prevented comprehension of the questionnaire (e.g., hearing impairment, mental disorders) (*n* = 9).

### 2.4. Data Collectiom

Ten trained interviewers conducted face-to-face interviews before or after participants’ consultation with health professionals in an individual, reserved room to have no external influence on the responses [47]. The data were collected between April and October 2019. The average time of the interviews was 18 minutes (±5 min, 9–55 min).

### 2.5. Outcome Variable: Leisure-Time Physical Activity (LTPA)

LTPA in a usual week was measured with the leisure module of the long version of the International Physical Activity Questionnaire (IPAQ) [48,49,50]. Participants self-reported the weekly frequency and the average daily volume of their walking and moderate- or vigorous-intensity PA. The minutes per week (min/week) for each variable and intensity were obtained by multiplying weekly frequency by the average daily volume of the PA types/intensities. Moderate-to-vigorous PA (MVPA) was measured by adding the minutes per week of moderate (excluded walking) and vigorous PA and multiplying the vigorous PA by two [48,50].

The LTPA was classified in two qualitative nominal variables, weekly walking and weekly MVPA, creating four outcomes: 1—some walking: ≥10 min/week; 2—high walking: ≥150 min/week; 3—some MVPA: ≥10 min/week; and 4—high MVPA: ≥150 min/week, according to WHO global recommendations [2].

These outcomes were chosen because studies have shown that some POS may affect LTPA differently according to PA type and cut-off points [15,16,23,31,51].

### 2.6. Predictors Variables: Exposure to Public Open Spaces (POS)

For this study, in the public health context, we conceptually understand POS as “spaces within the urban environment that are readily and freely accessible to the wider community, regardless of size, design or physical features and are intended primarily for amenity or recreation purposes—whether active or passive” [16]. The selected POS, locally known as sports and leisure centers, are free of charge and fully available for the local community. These POS are fenced and have PA structures (indoor sports courts, outdoor courts, fitness station, playgrounds, walking paths) where supervised PA and sports are provided (fitness classes, dance, soccer, volleyball, basketball) for children, adolescents, adults, and older adults in community-based PA programs (Active City, Healthy City Program) [29,32,34].

The POS are mainly located close to the BHUs in the urban area (two BHUs (13%) are on the same POS land, ten BHUs (67%) are located between 75 and 800 m from a POS, and only three BHUs (20%) are farther than 1500 m from a POS). The proximity between BHU and POS is an important characteristic to be considered for this study since one of the actions taken by the city administration to promote PA at the community level includes the PA counseling provided by BHU health professionals to encourage patients to use these POS [13,34] (Figure 1A).

There is 14 POS unequally distributed in the urban area (Figure 1A). They were personally visited and georeferenced with a global positioning system device (GPSMAP 64 s, Garmin, Taiwan, China). The geographic coordinates were checked with Google Earth Pro. We assessed four predictors of POS exposure, presence, distance, usage, and frequency of usage, as we describe below.

#### 2.6.1. Presence of Public Open Spaces (POS)

The participants’ addresses were collected during the interviews, and the residences were georeferenced using a geographical information system (GIS). A 500-m network buffer was created around the participants’ residences [52] using ArcGIS software (v 10.3 for Desktop, Environmental Systems Research Institute, Redlands, CA, USA) to identify the POS contained in the buffers (Figure 1 B, C). For analytical purposes, the POS presence was categorized as no (coded 0) or yes (coded 1 ≥ 1 POS). This distance can be reached in roughly five to ten minutes of walking and is frequently assessed in similar studies [15,31]. Moreover, as the distance to POS is a commonly reported PA barrier, 500 m might be adequate for our sample [53,54].

#### 2.6.2. Distance from Public Open Spaces (POS)

The shortest distance between a home and the nearest POS was calculated in meters with GIS considering the street network (Figure 1D). For inferences, this variable was ranked, stratified, and categorized into tertiles: 1st tertile: < 1156 m; 2nd tertile: 1156–2161 m; 3rd tertile: ≥ 2162 m.

#### 2.6.3. Public Open Spaces (POS) Usage

The POS usage was self-reported and evaluated with the question: *During the last year (12 months), did you use any POS (Sports and Leisure Centers) to realize PA*? (coded 0 for no or 1 for yes). This question has been used in similar studies, translated into Portuguese, adapted to a local context, and applied in studies conducted in Latin America [35,51,55,56]. This variable has shown adequate reliability (≥83% agreement, kappa ≥ 0.65; *p* < 0.001) [55].

#### 2.6.4. Frequency of Public Open Spaces (POS) Usage

The frequency of usage was evaluated when a positive answer was given to the previous question, and it was identified with nine response options: a few times a year, a few times a month, and once to seven times a week. This variable was classified on an ordinal scale: (1) no usage (code: 0); (2) usage a few times a year/month (code: 1); (3) usage once a week (code: 2) and (4) usage ≥2 times/week (code: 3) [51,55,56,57].

The type of activity undertaken was evaluated when the respondent was a POS user. The participants reported their main reason for using the POS with twelve response options (taking the children, dog walking, stretching, walking, running, cycling, skateboarding/skating, fitness classes, indoor activities, sports, exercise in fitness zone, and others) [51]. This variable was used only for descriptive purposes.

### 2.7. Possible Confounders Variables

Based on the literature review, the possible confounding variables included sociodemographic characteristics and chronic diseases [11,12,20,55].

#### 2.7.1. Sociodemographic Characteristics

Sex was observed, age was grouped into two categories (young adults: 18–39.9 years, middle-aged adults: 40–59.9 years), and marital status was assessed in three categories (single, married or stable relationship, divorced or widowed) and categorized into single (single, divorced, widowed) or married (married, stable relationship). Skin color was self-reported in five categories (white, black, yellow, brown, indigenous) and grouped as white and nonwhite (other categories). These variables were measured according to the Brazilian Institute of Geography and Statistics [40]. The participants’ socioeconomic status (SES) was assessed with a standard questionnaire that considers the possession of household appliances, having a housekeeper, the education level of the family head, and public services at the residence (water and paved street). The SES was classified into seven categories (A1–highest, then A2, B1, B2, C, D, E) [58] and categorized into low (C + D + E) or high (A + B) SES for analytical purposes.

#### 2.7.2. Chronic Diseases

Overweight was established by body mass index (kg/m^2^-body mass and height self-reported) [59] and classified into two categories: ≤24.9 kg/m^2^ and ≥25.0 kg/m^2^ [60]. The presence of chronic diseases was dichotomized (no, yes) according to a self-reported medical diagnosis of hypertension, diabetes, dyslipidemia, coronary artery disease, circulatory diseases, or lung diseases following the Surveillance System for Risk Factors and Protection for Chronic diseases compiled by telephone survey [61]. Finally, the number of chronic diseases was calculated by summing overweight and other chronic diseases and classified into three categories: 0, 1, or ≥2. This variable has been used in other studies [55] and may be relevant as health care users have higher chronic disease prevalence [8,9,10], which may impair their POS usage and LTPA. On the other hand, people with chronic diseases are more exposed to PA counseling by health care professionals, which could help them change their behavior and start using the POS [13,14].

### 2.8. Data Quality Control

Data quality control was assured through seven steps. First, all interviewers received 20 h theoretical and practical training on technical procedures for interviewing (approaching participants, recording losses and refusals, applying questionnaires, and coding forms) based on the instruction manual prepared by the core project team. The interviewers strictly followed all these procedures; they were blinded to the objectives and hypotheses of the investigation and supervised by a field coordinator. Second, a pilot study was conducted on a random sample of 81 participants from three BHUs to test the data collection procedures and the comprehension of the questions translated from other studies and adapted to the local context. Third, all the pilot study participants were interviewed at an interval of 7 to 10 days to analyze the temporal stability of the main variables of the study. Fourth, data entry was conducted by the field coordinator in EpiData software (EpiData Association, Odense, Denmark). Fifth, data cleaning used exploratory analysis in SPSS software (v. 26.0, IBM SPSS Statistics, Armonk, NY, USA) to identify possible typing errors in data entry for each variable, detect outliers, and verify all variable distributions. Sixth, each variable outlier was personally checked in the questionnaire and manually corrected in the database. Finally, all analyses of spatial data were performed twice, reviewed, and validated by one of the authors, a specialist in GIS with expertise in built environments and PA (A.A.S.L.).

### 2.9. Data Analysis

Quantitative data were analyzed using descriptive statistics (average, standard deviation, frequency distribution). The prevalence of walking and MVPA were described among the categories of POS exposure, and their associations were analyzed by multilevel Poisson regression [62] to account for the clustering at the Basic Health Unit level as random effects. All potential confounders (sociodemographic characteristics and number of chronic diseases) that presented a *p*-value < 0.20 in the bivariate analysis were included for adjustment into the multilevel model Poisson regression as fixed effects, and the variance partition coefficient (VPC) was calculated for each combination of outcomes and exposure, allowing for identifying the proportion of the variance in LTPA attributed to the BHU [63,64]. All analyses were performed using the STATA software (v. 16, StataCorp, College Station, TX, USA) and at a 5% significance level.

## 3. Results

A total of 947 BHU users were invited to participate (≥18 years old), the refusal rate was 16% (*n* = 147), and 2% did not finish the interviews (*n* = 21), which resulted in 779 respondents. However, 18% of them (*n* = 141) were excluded because they were out of the age analyzed in this study (≥60 years old). Among the remaining 638 eligible adults, 4% (*n* = 23) did not provide a residential address, and we could not geolocate their residences. Thus, the final analytical sample for this study was 615 participants (Figure 1A). The original (*n* = 779) and the analytical (*n* = 615) samples were similar according to the proportions of participants by sex, economic level, chronic diseases, and LTPA (walking and MVPA) (*p* > 0.05).

Most participants were women (73.5%), between 18 and 39.9 years (54.3%), married (64.2%), white (73.5%), and from a low economic level (69.9%) (Table 1). Regarding the chronic diseases, 67.8% were overweight, 28.5% reported hypertension, 13.3% circulatory disease, 12.4% dyslipidemia, 12.2% diabetes, 5.7% lung disease, and 4.9% coronary artery disease. About 77% of the participants had ≥ 1 chronic disease (Table 1).

A quarter of the participants (24.7%) reported walking during leisure time (≥10 min/week), and 11.7% walked ≥ 150 min/week. About 21% reported some MVPA (≥10 min/week), and 13.8% practiced MVPA ≥ 150 min/week (Table 1).

The average distance from residences to POS was 1880 ± 1494 m (median: 1589 m; minimum: 78 m; maximum: 10,279 m; interquartile range: 1469 m). POS were available for 10.2% of the participants, 15.8% reported using the places, and 8.1% of these had usage of ≥ 2 times/week (Table 1).

Among the POS users (15.8% of the sample), 52% visit the places ≥2 times/week. The main reasons for POS usage were engaging in PA classes (45%), sports (19%), and walking (14%) (Figure 2).

In the bivariate associations between possible confounders and LTPA (Table 2), age (PR: 1.59, 95%CI: 1.20–2.10) and number of chronic diseases (≥2, PR: 1.70, 95%CI: 1.17–2.45) were positively associated with walking ≥10 min/week. Age (PR: 1.87, 95%CI: 1.19–2.92), economic level (PR: 1.57, 95%CI: 1.01–2.43), and number of chronic diseases (≥2, PR: 1.98, 95%CI: 1.07–3.64) were associated with walking ≥150 min/week (Table 2). Regarding MVPA, female sex was inversely associated with ≥10 min/week (PR: 0.62, 95%CI: 0.46–0.85), while economic level (PR: 1.53, 95%CI: 1.13–2.07) was positively associated. Only economic level was positively associated with MVPA ≥150 min/week (PR: 1.63, 95%CI: 1.10–2.42) (Table 2).

In the bivariate analysis of POS exposure and LTPA, the usage (PR: 1.54; 95%CI: 1.04–2.23) and the higher frequency of POS usage (≥2 times/week, PR: 2.20; 95%CI: 1.39–3.32) showed significant positive associations with walking ≥10 min/week (Table 3). Regarding walking ≥150 min/week, POS presence was marginally associated (PR: 1.75; 95%CI: 0.90–3.14), while a higher frequency of POS usage showed a significant positive association (≥2 times/week, PR: 2.04; 95%CI: 1.01–3.73) (Table 3). POS usage and a higher frequency of usage were positively associated with both MVPA outcomes (*p* < 0.001) (Table 4).

After adjustment for possible confounders, the POS usage (PR: 1.52; 95% CI: 1.02–2.21) and the frequency of usage (≥2 times/week, PR: 2.01; 95%CI: 1.28–3.45) remained associated with walking ≥10 min/week. The POS presence (PR: 1.79; 95%CI: 0.92–3.21) and frequency of usage ≥2 times/week (PR: 1.85; 95%CI: 0.92–3.41) also maintained marginally positive associations with walking ≥150 min/week (Table 3). In the analysis of MVPA, the usage and higher frequency of POS usage remained positively associated with both outcomes (*p* < 0.001) (Table 4). POS users are three times more likely to meet MVPA recommendations for leisure time than nonusers (≥150 min/week, PR: 3.03; 95%CI: 1.92–4.74). This probability increases among those who use the POS ≥2 times/week (PR: 3.81; 95%CI: 2.19–6.38).

## 4. Discussion

To the best of our knowledge, this is the first study that examined the associations between exposure to POS and the LTPA of adult primary health care users. The quantitative approach and the geoprocessing analysis in a representative sample of adults allowed us to verify important contextual characteristics of POS in a medium-sized city located close to BHUs for PA promotion at the community level. Four possible POS exposure predictors (presence, distance, usage, and frequency of usage) were simultaneously evaluated, which is uncommon in studies that utilized objective measures to examine the associations between accessibility (presence, density, distance) and LTPA [15,22,31]. Furthermore, the variables were measured with proper and standardized procedures, allowing for comparison between studies.

Undertaking the analysis of adult primary health care users in Brazil can help with understanding how POS exposure can affect LTPA in this population [9,10,37,38]. Most of the evidence for the association between POS and LTPA comes from studies conducted with apparently healthy participants in high-income countries [11,12,16,17,19,21]. In Brazil, the population that depends exclusively on the public health system for primary care has a low economic level, a low LTPA level, and many chronic diseases [9,10]. Enabling free access to POS in the neighborhood with suitable-quality classes and structures could stimulate and facilitate people’s engagement in LTPA to obtain health benefits [11,12]. From an international perspective, exploring these variables is important so that policymakers can develop public policies and plan actions for the construction, maintenance, and expansion of access to POS to promote LTPA at the population level in medium-sized cities, which can reduce the burden of several risk factors for noncommunicable diseases [5,6,7]. For example, it is estimated that in Latin America and Europe, a large part of the population resides in cities with fewer than 500,000 inhabitants [65].

The presence and usage of POS ≥2 times/week marginally increased the likelihood of walking ≥150 min/week by 79% and 85%, respectively. Most of the research conducted in countries of different incomes has found inconsistent or null associations between the presence of recreational and sports facilities and leisure walking [17,19]. When significant associations were found, the magnitude varied between 30 and 66% in low- and middle-income countries [31,66] and between 43 and 70% in high-income countries [67,68]. This variability can be explained because only POS presence around residences may not be enough to increase walking if these places are not attractive or suitable for this PA (safe with a walking path) [11,12,16,22,31]. In the present study, the presence of walking paths in the POS may partially explain the strength of association found, as walking and dog walking were two of the main reasons for usage (15%, Figure 2). Leisure-time walking, even in small doses, positively affects health and can reduce the risk of mortality by 40% [69].

Our results showed that POS usage increased, significantly, the likelihood of walking ≥ 10 min/week by 52% and increased by 2.7 and 3 times the MVPA ≥10 min/week and ≥150 min/week, respectively. Few studies have examined the relationship between POS usage and LTPA, especially at higher intensity [70,71,72,73]. In general, the positive and significant associations [71,72,73] have shown that using these places can increase leisure walking by 17–390% and MVPA by 55–420% [72]. The wide magnitude of variability can be explained by the contextual characteristics of the countries and POS settings. For instance, Mackenbach et al. [72] analyzed the usage of formal recreational facilities (public and private courts, parks, and gyms) in European adults and reported increases of 17% and 55% in walking and MVPA, respectively. On the other hand, Salvo et al. [73] examined the usage of formal and informal POS by adults from three Latin American countries (Brazil, Colombia, and Mexico) and reported increases of 4.9 and 5.2 times in the odds of walking and MVPA, respectively, taking the Mexican population as an example. An essential contextual characteristic of the latter study was the high usage of informal POS in Colombia, such as in shopping malls (54%) [73].

The differences mentioned above reinforce the relevance of analyzing different locations and contexts when examining the possible impact of POS usage on LTPA [73]. For example, while less than 16% of our sample reported using POS, in other Latin American cities, the POS usage (parks, plazas, fitness zone, cycling or walking paths, outdoor courts) is higher (61–75%) [53,55,73]. In part, the difference can be explained by the profiles of primary health care users in the Brazilian Unified Health System, who are characterized by limited access, time available, and interest in the places [11,12,74]. Most of the study participants were female, of low income, and married; had a chronic disease; and were insufficiently active in leisure time: characteristics that highlight the importance of POS with features and community programs to promote LTPA [36,37].

Even with a lower POS usage prevalence, the magnitudes of the associations were similar or superior to those found in Latin American adults [73], perhaps because the main reasons for visiting POS (88%) were related to MVPA practice (PA classes, 45%; sports, 19%; walking, 15%; exercise in a fitness zone, 5%; running, 2%; and fighting classes, 2%) [75] (Figure 2). Moreover, systematic observations conducted in these POS showed that 91% of the users practiced some PA, and 52% participated in MVPA [29). As the POS have sports courts and offers supervised PA classes, these characteristics may have contributed positively to MVPA [32,34].

Higher POS usage (≥2 times/week) increased the likelihood of walking and practicing MVPA by approximately two and four times, respectively. The association between POS usage and LTPA is poorly studied, although the positive association is consistent among studies [51,56]. Our results corroborate those findings, but it is nevertheless important to highlight the differences in associations according to POS and LTPA types. Some studies that evaluated the frequency of park usage found an increase in leisure walking of 26–560% with higher weekly visitations [51,56]. In our study, the greatest magnitude was observed for MVPA, which can be explained by the high participation in PA classes and sports (Figure 2). Some authors have suggested that parks can be better facilitators of walking while recreational centers might have a more significant impact on MVPA [23].

Contradicting our expectations, POS presence and proximity were not associated with most PA outcomes. Other studies have also reported null or inconsistent associations, except for positive associations between the presence and use of recreational facilities in low- and middle-income countries [17,19,21]. The lack of association may be related to intrapersonal barriers to practicing LTPA among Brazilian adults, regardless of POS presence or proximity (lack of knowledge of PA classes and activities available at the POS; lack of motivation and time; higher prevalence of chronic diseases) [23,54,74]. Moreover, although the POS assessed have common characteristics, the structure quality and activities offered (type, quantity, and variety) differ between places and could help to explain the lack of associations.

Some limitations should be considered for an adequate interpretation of the results. First, the sample is not representative of the population or the entire primary health care network: it is limited to adult BHU users in the urban area of a medium-sized city in southern Brazil. Second, the objective measurement of POS presence and distance by geoprocessing did not consider the perception of neighborhood aspects (proximity and safety) and POS quality (safety, structure, and services). Third, we evaluated exposure to only one type of POS, but the city has others (parks, woods, plazas, free spaces), and the results cannot be extrapolated to them. Fourth, the recall bias inherent in the self-reported measure of PA may have negatively affected the associations between the exposure variables and LTPA [47,76]. In fact, studies have shown a positive association between the presence and distance of POS with objectively measured PA [77]. However, it is important to highlight that the subjective measurement of PA makes it possible to identify the domains in which activities are performed (e.g., commuting, leisure), which can significantly differ from the total PA measured by objective methods [76,78,79]. Finally, the cross-sectional study design does not establish causality between POS exposure and LTPA outcomes.

## 5. Conclusions

Only the usage of POS is positively associated with walking and MVPA in the leisure time of adult primary health care users in Brazil.

Our results are relevant and should be considered for guiding policymakers and health care professionals about PA promotion in primary health care. Policymakers could plan and implement PA-oriented programs at BHUs at least once a week, and health care professionals could encourage and advise their patients to attend the community-based PA programs offered at the POS. The construction and maintenance of other safe POS with quality structures that are adequate for the practice of different PA can facilitate access and encourage other activities for the population that uses primary health care.

Future studies could advance the analysis and evaluate the effect of health care professionals’ counseling for POS usage and its influence on the LTPA levels of adults in primary health care. It is essential to evaluate the interaction effects of POS quality perception, usage, and LTPA. It is also relevant to explore the direct and indirect effects of counseling, POS presence, and proximity on LTPA to better understand the inconsistent associations observed in the literature.

## Figures and Tables

**Figure 1 ijerph-19-08355-f001:**
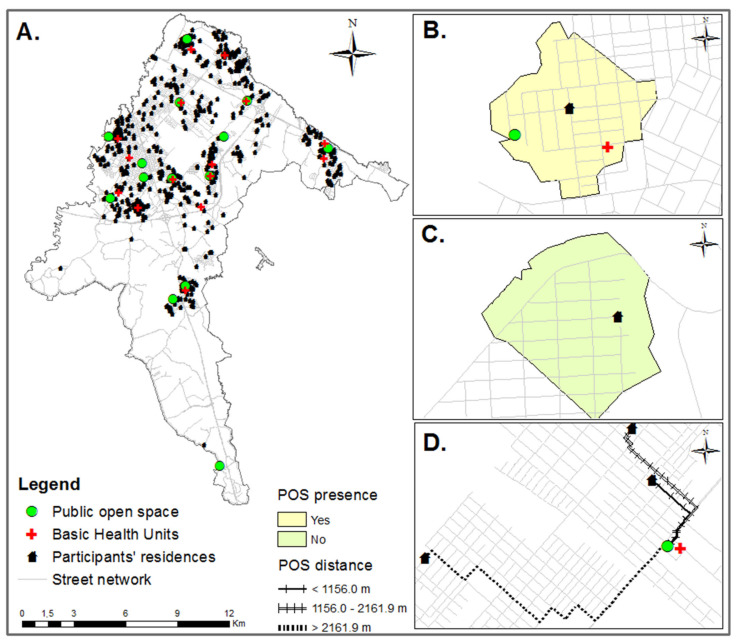
(**A**): Public open space (POS) (*n* = 14), Basic Health Units (*n* = 15), and the participants’ residence (*n* = 615) spatial locations; (**B**): POS present (500-m street network buffer); (**C**): POS absent; (**D**): shortest distance between residence and POS (tertiles in meters). São José dos Pinhais, Paraná, Southern Brazil, 2019.

**Figure 2 ijerph-19-08355-f002:**
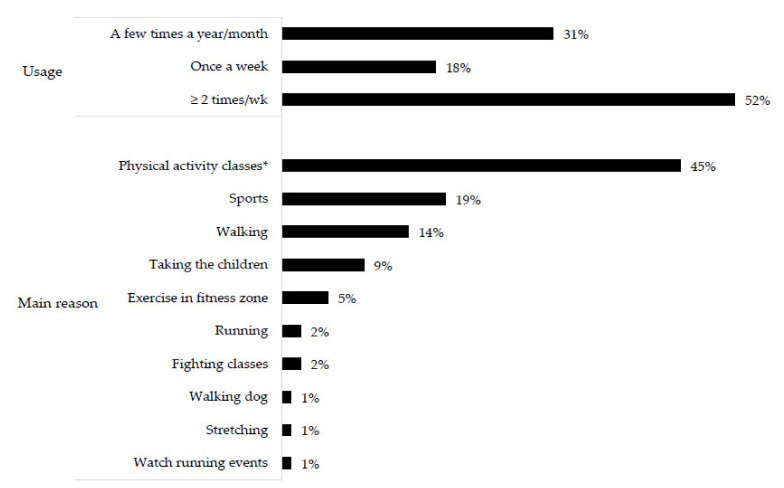
Descriptions of the usage of public open spaces by adults in primary health care. São José dos Pinhais, Southern Brazil, 2019 (*n* = 97) (* aerobics, Zumba, and functional exercises).

**Table 1 ijerph-19-08355-t001:** Sample characteristics of adults in primary health care. São José dos Pinhais, Southern Brazil, 2019 (*n* = 615).

Variable	Category	*n*	%	Average	S.D.
Sociodemographic					
Sex	Male	163	26.5	-	-
	Female	452	73.5	-	-
Age (years)	18–39.9	334	54.3	38.4	12.4
	40–59.9	281	45.7		
Marital status	Single	220	35.8	-	-
	Married	395	64.2	-	-
Skin color	White	450	73.5	-	-
	Non-white	162	26.5	-	-
Economic level	Low	430	69.9	-	-
	High	185	30.1	-	-
Chronic diseases					
Overweight	No	196	32.2	27.7	5.2
	Yes	413	67.8		
Hypertension	No	440	71.5	-	-
	Yes	175	28.5	-	-
Circulatory disease	No	533	86.7	-	-
	Yes	82	13.3	-	-
Dyslipidemia	No	539	87.6	-	-
	Yes	76	12.4	-	-
Diabetes	No	540	87.8	-	-
	Yes	75	12.2	-	-
Lung disease	No	580	94.3	-	-
	Yes	35	5.7	-	-
Coronary artery disease	No	585	95.1	-	-
	Yes	30	4.9	-	-
Number of chronic diseases *	0	141	22.9	1.4	1.2
	1	236	38.4		
	≥2	238	38.7		
Leisure time physical activity					
Walking	0–9 min/week	463	75.3	43.2	106.9
	≥10 min/week	152	24.7		
	0–149 min/week	543	88.3		
	≥150 min/week	72	11.7		
MVPA	0–9 min/week	484	78.7	72.4	215.8
	≥10 min/week	131	21.3		
	0–149 min/week	530	86.2		
	≥150 min/week	85	13.8		
Public open space exposure **					
Presence ***	No	552	89.8	-	-
	Yes	63	10.2	-	-
Distance	<1156 m	201	33.3	1880.7	1494.8
	1156–2161 m	202	33.4		
	≥2162 m (3rd tertile)	201	33.3		
Usage	No	518	84.2	-	-
	Yes	97	15.8	-	-
Frequency of usage	No use	518	84.2	-	-
	A few times a year/month	30	4.9	-	-
	Once a week	17	2.8	-	-
	≥2 times/week	50	8.1	-	-

S.D.: standard deviation, min/week: minutes per week, times/week: times per week, MVPA: moderate-to-vigorous physical activity, * Sum of chronic diseases, ** Sports and Leisure Centers, *** 500-m network buffer.

**Table 2 ijerph-19-08355-t002:** The bivariate associations between sociodemographic characteristics, number of chronic diseases, and the leisure-time physical activity of adults in primary health care. São José dos Pinhais, Southern Brazil, 2019 (*n* = 615).

	Walking ≥ 10 min/week				Walking ≥ 150 min/week				MVPA ≥ 10 min/week				MVPA ≥ 150 min/week			
	%	PR	95% CI	*p*	%	PR	95% CI	*p*	%	PR	95% CI	*p*	%	PR	95% CI	*p*
Sex																
Male	24.5	1		0.952 ^h^	12.9	1		0.585 ^h^	29.4	1		**0.003 ^h^**	17.2	1		0.145 ^h^
Female	24.8	1.01	0.74–1.38		11.3	0.88	0.54–1.41		18.4	**0.62**	0.46–0.85		12.0	0.73	0.48–1.11	
Age (years)																
18–39.9	19.5	1		**<0.001 ^h^**	8.4	1		**0.006 ^h^**	22.5	1		0.447 ^h^	14.4	1		0.667 ^h^
40–59.9	31.0	**1.59**	1.20–2.10		15.7	**1.87**	1.19–2.92		19.9	0.89	0.65–1.21		13.2	0.92	0.61–1.36	
Marital status																
Single	23.2	1		0.513 ^h^	10.9	1		0.647 ^h^	22.7	1		0.518 ^h^	14.5	1		0.697 ^h^
Married	25.6	1.10	0.82–1.48		12.2	1.11	0.70–1.77		20.5	0.90	0.66–1.23		13.4	0.92	0.61–1.39	
Skin color																
White	25.3	1		0.531 ^h^	11.8	1		0.987 ^h^	21.6	1		0.753 ^h^	14.0	1		0.895 ^h^
Non-white	22.8	0.90	0.65–1.25		11.7	1.00	0.61–1.63		20.4	0.95	0.66–1.34		13.6	0.97	0.62–1.52	
Economic level																
Low	22.6	1		0.055 ^h^	10.0	1		**0.044 ^h^**	18.4	1		**0.006 ^h^**	11.6	1		**0.016 ^h^**
High	29.7	1.32	0.99–1.75		15.7	**1.57**	1.01–2.43		28.1	**1.53**	1.13–2.07		18.9	**1.63**	1.10–2.42	
Number of chronic diseases *																
0	20.6	1		**0.001 ^t^**	8.5	1		**0.012 ^t^**	26.2	1		0.210 ^t^	15.6	1		0.836 ^t^
1	16.9	0.82	0.54–1.27		8.5	1.00	0.50–1.97		18.6	0.71	0.48–1.04		12.3	0.79	0.47–1.32	
≥2	34.9	**1.70**	1.17–2.45		16.8	**1.98**	1.07–3.64		21.0	0.80	0.55–1.16		14.3	0.92	0.56–1.50	

MVPA: moderate-to-vigorous physical activity, PR: prevalence ratio, 95% CI: 95% confidence interval, min/week: minutes per week, * Sum of chronic diseases, ^h^ Wald test for heterogeneity, ^t^ Wald test for trend.

**Table 3 ijerph-19-08355-t003:** The multilevel associations between exposure to public open space and walking in the leisure time of adults in primary health care. São José dos Pinhais, Southern Brazil, 2019 (*n* = 615).

	Walking ≥ 10 min/week							Walking ≥ 150 min/week						
	Bivariate Analysis				Multivariate Analysis *			Bivariate Analysis				Multivariate Analysis *		
	%	PR	95% CI	*p*	PR	95% CI	*p*	%	PR	95% CI	*p*	PR	95% CI	*p*
Presence														
No	23.9	1		0.237 ^h^	-	-	-	10.9	1		**0.076 ^h^**	1		**0.065 ^h^**
Yes	31.7	1.33	0.80–2.10		-	-	-	19.0	**1.75**	0.90–3.14		**1.79**	0.92–3.21	
		0.011 (0.003) #			-	-	-		0.000 (0.000) #			0.000 (0.000) #		
Distance														
<1156 m	26.4	1		0.699 ^t^	-	-	-	13.9	1		0.189 ^t^	-	-	-
1156–2161 m	22.3	0.85	0.57–1.26		-	-	-	11.4	0.82	0.47–1.42		-	-	-
≥2162 m	24.4	0.93	0.63–1.40		-	-	-	9.5	0.68	0.37–1.21		-	-	-
		0.003 (0.001) #			-	-	-		0.000 (0.000) #			-	-	-
Usage														
No	22.8	1		**0.027 ^h^**	1		**0.031 ^h^**	10.8	1		0.136 ^h^	-	-	-
Yes	35.1	**1.54**	1.04–2.23		**1.52**	1.02–2.21		16.5	1.53	0.85–2.59		-	-	-
		0.010 (0.003) #			0.000 (0.000) #				0.000 (0.000) #			-	-	-
Frequency of usage														
No use	22.8	1		**<0.001 ^t^**	1		**<0.001 ^t^**	10.8	1		**0.047 ^t^**			**0.076 ^t^**
A few times a year/month	20.0	0.88	0.34–1.84		0.95	0.37–1.97		10.0	0.93	0.23–2.50		1.02	0.25–2.79	
Once a week	17.6	0.77	0.19–2.05		0.84	0.21–2.23		11.8	1.09	0.18–3.49		1.14	0.19–3.69	
≥2 times/week	50.0	**2.20**	1.39–3.32		**2.01**	1.28–3.05		22.0	**2.04**	1.01–3.73		**1.85**	0.92–3.41	
		0.007 (0.002) #			0.000 (0.000) #				0.000 (0.000) #			0.000 (0.000) #		

PR: prevalence ratio, 95% CI: 95% confidence interval, times/week: times per week, *: adjusted (included as fixed effects) for variables with a *p*-value < 0.20 in the bivariate association with walking ≥10 min/week and ≥150 min/week (age, economic level, number of chronic diseases), ^h^: Wald test for heterogeneity, ^t^: Wald test for trend; #: values for intercept (random effects) and variance partition coefficient (VPC).

**Table 4 ijerph-19-08355-t004:** The multilevel associations between exposure to public open space and the MVPA in the leisure time of adults in primary health care. São José dos Pinhais, Southern Brazil, 2019 (*n* = 615).

	MVPA ≥ 10 min/week							MVPA ≥ 150 min/week						
	Bivariate Analysis				Multivariate Analysis *			Bivariate Analysis				Multivariate Analysis ^§^		
	%	PR	95% CI	*p*	PR	95% CI	*p*	%	PR	95% CI	*p*	PR	95% CI	*p*
Presence														
No	20.7	1		0.279 ^h^	-	-	-	13.9	1		0.800 ^h^	-	-	-
Yes	27.0	1.34	0.77–2.20		-	-	-	12.7	0.91	0.40–1.80		-	-	-
		0.019 (0.006) #			-	-	-		0.000 (0.000) #			-	-	-
Distance														
<1156 m	20.9	1		0.797 ^t^	-	-	-	13.9	1		0.893 ^t^	-	-	-
1156–2161 m	19.3	0.92	0.63–1.36		-	-	-	12.9	0.92	0.53–1.58		-	-	-
≥2162 m	22.9	1.06	0.67–1.65		-	-	-	14.4	1.04	0.59–1.75		-	-	-
		0.017 (0.005) #			-	-	-		0.000 (0.000) #			-	-	-
Usage														
No	16.8	1		**<0.001 ^h^**	1		**<0.001 ^h^**	10.4	1		**<0.001 ^h^**	1		**<0.001 ^h^**
Yes	45.4	**2.85**	1.96–4.13		**2.73**	1.86–3.97		32.0	**3.13**	1.97–4.89		**3.03**	1.92–4.74	-
		0.054 (0.016) #			0.047 (0.014) #				0.026 (0.008) #			0.008 (0.002) #		
Frequency of usage														
No use	16.8	1		**<0.001 ^t^**	1		**<0.001 ^t^**	10.4	1		**<0.001 ^t^**	1		**<0.001 ^t^**
A few times a year/month	33.3	**2.09**	1.01–3.89		1.77	0.84–3.35		20.0	1.95	0.75–4.24		1.76	0.67–3.85	
Once a week	47.1	**2.87**	1.27–5.96		**2.58**	1.14–5.07		35.3	**3.45**	1.32–7.48		**3.15**	1.21–6.82	
≥2 times/week	52.0	**3.31**	2.06–5.16		**3.45**	2.14–5.42		38.0	**3.71**	2.12–6.21		**3.81**	2.19–6.38	
		0.056 (0.0017) #			0.046 (0.014) #				0.024 (0.007) #			0.000 (0.000) #		

MVPA: moderate-to-vigorous physical activity, PR: prevalence ratio, 95% CI: 95% confidence interval, times/week: times per week, *: adjusted (included as fixed effects) for variables with a *p*-value <0.20 in the bivariate association with MVPA ≥10 min/week (sex, economic level, and number of chronic diseases), ^§^: adjusted (included as fixed effects) for variables with a *p*-value < 0.20 in the bivariate association with MVPA ≥150 min/week (sex and economic level), ^h^: Wald test for heterogeneity, ^t^: Wald test for trend; #: values for intercept (random effects) and variance partition coefficient (VPC).

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
