# Peer review of "Exposure to Public Open Spaces and Leisure-Time Physical Activity: An Analysis of Adults in Primary Health Care in Brazil"

_ijerph, 2022, doi:10.3390/ijerph19148355_

Round 1
Reviewer 1 Report
In this manuscript, the authors aim to investigate the relationship between public spaces and the level of leisure-time physical activity, using data collected from self-administered surveys by hospital users. The authors’ investigation may contribute to the understanding of the effect of spatial availability on how people conduct physical activity (PA).
In general, the research topic is intriguing. However, the authors must input significant efforts to solve many issues throughout the manuscript including the incomplete introduction, the distracted literature review, the insufficient methods explained, the need for information in analyses and discussion, and the need for in-depth study implications. Those issues will require improvements to prevent confusion and increase the study logic.
I would strongly suggest the authors have editorial assistance throughout the entire manuscript to increase its readability. As in the current manuscript, many sentences appear fragmented or very confusing. They are obviously under development. This issue adversely affects the quality of the manuscript which makes this manuscript looks like the authors’ first draft. I sincerely suggest the authors put a good effort to reorganize and recreate the thread of the manuscript.
I strongly suggest the authors take another check to see if the information in the sections is linked to the focus of this study. There is much information that seems to be distracted from the focus of this study, such as the long description of health systems (L47-L72), other countries’ area comparisons (L114-L121), and detailed IRB description(L122-L125). Conversely, the real needed pieces of literature are barely mentioned which presents a deficiency in this important section. Obviously, the information needs to be carefully filtered to improve its readability.
I also suggest the authors develop a complete study argument that links the pieces together to form the thread of argument. Right now, the argument seems too general and the link to this current study is weak.
Author Response
Caro revisor,
Obrigado por avaliar o artigo intitulado "Exposição a espaços públicos abertos e atividade física no lazer: uma análise de adultos na Atenção Primária à Saúde no Brasil" (ID IJERPH 1731794), submetido em 02 de maio de 2022.
Os comentários e sugestões certamente contribuíram para o aprimoramento do manuscrito. Em anexo estão as respostas "ponto a ponto" a cada comentário. As correções e alterações foram feitas no papel anexo e destacados em amarelo. Acreditamos que todas as solicitações foram atendidas ou devidamente justificadas.
Enviamos uma cordial saudação e esperamos a decisão mais adequada.
Sinceramente.
Pelos autores.
Rogério César Fermino, Ph.D.

Reviewer 2 Report
Thank you for allowing me to review the original manuscript entitled Exposure to “Public Open Spaces and Leisure-Time Physical Activity: An Analysis of Adults in Primary Health Care in Brazil”. The topic is relevant, the manuscript is well-written and I have no major concerns in the methodology. I especially enjoy how the authors discuss their results from an international perspective, giving context-specific explanations. However, before considering it for publication there are some issues that, in my opinion, the authors should address:
General comments:
- I think the manuscript will benefit from a clear definition of POS. The literature is very weak about clear definition of open spaces; some include only open, some include gyms, some not…
- In the introduction, the authors could justify what is the advantage of evaluation all POS together, as one of the justifications for their analysis is that there are no many manuscripts analyzing POS together.
- Was there any differences between the original sample and the analytical sample?
- For the availability analysis, did you try other buffers (as sensitivity) besides 500-m?
- Regarding confounding variables. Did you check if the potential confounding variables had effect modification? I wonder if some of the potential confounders are not confounders at all, and might be mediators or even colliders. For example, to be a confounder, a variable has to: (1) be a risk factor of the disease (in this case, you checked that with previous models), (2) be associated with the exposure, and (3) not being in the causal pathway. I suggest the authors at least check the second point to consider potential confounders in the analysis.
- I’d suggest to use some multi-level structure in your model to account for the clustering of your participants into the BHU.
- I’d add to the limitations the measures of physical activity. In parks, there is evidence that this matters (see https://www.sciencedirect.com/science/article/abs/pii/S0277953615003160?via%3Dihub )
- Finally, after the review, I would suggest working on the format of the submission. Some tables are difficult to read, and there is some text disconnected from other parts.
Specific and minor comments:
Line 80 references repeated
Line 221 it seems some word is missing in the sentence
Author Response
Dear reviewer,
Thank you for evaluating the paper entitled "Exposure to public open spaces and leisure-time physical activity: an analysis of adults in Primary Health Care in Brazil" (ID IJERPH 1731794), submitted on May 02, 2022.
The comments and suggestions certainly contributed to the improvement of the manuscript. Attached are the "point-to-point" responses to each comment. Corrections and changes were made in the paper attached and highlighted in yellow. We believe that all requests have been met or duly justified.
We send a cordial greeting and hope for the most appropriate decision.
Sincerely.
By the authors.
Rogério César Fermino, Ph.D.

Reviewer 3 Report
The authors developed an interesting article for the reader of the International Journal of Environmental Research and Public Health, which added knowledge about association between the public open spaces and leisure-time physical activities in adults in primary health care in Brazil.
In general, the article is well structured but in my opinion it’s a little long. However, before its publication, I leave some comments that can be taken into account by the authors:
1 - The materials and methods are described allowing the replication of the study. The characteristics of the sample, all the instruments used, as well as the statistical tests are presented. However, some sections of the materials and methods could be written more succinctly.
2- [Line 152-158] I didn’t understand the participant selection methods used. Maybe it would be better to explain it another way.
3- Section 2.6: I think the pictures shown are not of scientific interest.
4- Section 2.6.1: In my opinion figure 2 would be better in the results.
5- [Line 229 and 241] Add the manufacturer and country of the equipment used.
6- Section 2.6.3 and 2.6.4: Why the use of these questions? Were previously validated? Id yes, add reference.
7- [Line 282] Add questionnaire reference.
8- Section 2.7.2: How was body mass index measured? Add reference to the instrument used. Or was it just asked to the participants?
9- [Line 337] “participate (318 yrs old)” I didn’t understand.
10- Table 2: Make the entire text of the two lines bold.
I would like to take this opportunity to congratulate the authors for their excellent research work. I wish you all the best and keep up the good work.
Author Response

(The authors gave the same response as above.)

Reviewer 4 Report
I would like to congratulate the authors for the quality and methodological rigor of this study. You have addressed a very important issue globally and especially in Latin America and Brazil. I think it is essential, as you have rightly pointed out, the role of health professionals in promoting physical activity and especially that in open spaces. I believe the social, cultural and scientific impact of this article is high.
Well done.
Author Response

(The authors gave the same response as above.)

Round 2
Reviewer 2 Report
The authors have addressed most of the comments in a correct manner. I would like to thank them for the clarity in the presentation of their changes.